# Application of Artificial Intelligence in the Assessment and Forecast of Avalanche Danger in the Ile Alatau Ridge

Viktor Blagovechshenskiy *, Akhmetkal Medeu, Tamara Gulyayeva, Vitaliy Zhdanov, Sandugash Ranova, Aidana Kamalbekova and Ulzhan Aldabergen

Institute of Geography and Water Safety, Almaty 050010, Kazakhstan
* Correspondence: victor.blagov@mail.ru; Tel.: +997-705-768-2378

**Abstract:** The assessment and forecast of avalanche danger are very important means of preventing avalanche fatalities, especially in recreational areas. The use of artificial intelligence methods for these purposes significantly increases the accuracy of avalanche forecasts. The purpose of this re-search was to improve the methods for assessing and forecasting avalanche danger in the Ile Alatau Ridge. To create a training sample, the data from three meteorological and two avalanche stations for the period from 2002 to 2022 were used. The following predictors were chosen: air temperature, snow cover depth, precipitation, and snowpack stability index. The subject of the assessment and forecasts was the level of avalanche danger, assessed on a five-point scale. The program Statistica StatSoft was used as a neurosimulator. When forecasting avalanche danger, the predictive values of air temperature and precipitation, obtained from numerical weather forecast models, were used. The model correctly assessed the current level of avalanche danger in 90% of cases. The forecast of avalanche danger was justified in 80% of cases. The artificial intelligence program helped the avalanche forecaster to improve the forecast quality. This method is currently being used for compiling an avalanche bulletin for two river basins in the Ile Alatau.

**Keywords:** artificial intelligence; artificial neural network; avalanche danger forecast; avalanche danger assessment; Northern Tien Shan

## 1. Introduction

Snow avalanches pose a great danger to people and economic objects. They are widely distributed in many mountainous regions of the world. Avalanches create a particularly big problem for ski resorts and travel companies organizing ski tours and freerides. Every year, an average of 90 skiers and tourists die in avalanches in the European Alps [1]. In Kazakhstan, over the past 71 years, their number has amounted to 98 victims [2]. Currently, most of the avalanche incidents occur in unprotected areas among fans of extreme winter sports—freeriders and ski tourists [1]. Avalanche forecast and avalanche danger warning play a key role in preventing such accidents. In Switzerland, the avalanche danger forecast and avalanche bulletin are compiled at the federal level by the Snow and Avalanches Research Institute (SLF) [3–6], in France by the Meteo France meteorological service [7,8], in the USA and Canada by regional avalanche centers [9–13]. In addition, each ski resort has its own avalanche service that issues a local avalanche bulletin with an avalanche danger forecast. In Kazakhstan, avalanche danger warnings are issued by the regional branches of the Kazakhstan Hydrometeorological Service [14,15]. This warning is sent to the Ministry of Emergency Situations, which alerts the population and carries out avalanche prevention measures: closure of the territory and artificial avalanching. At the Shymbulak and Akbulak ski resorts in the Ile Alatau, avalanche danger assessment and avalanche prevention measures are carried out by the ski patrol services.

The first attempts to forecast avalanches were made in Switzerland and the USSR in the 1930s [16,17]. They were widely developed in the second half of the 20th century.

With the development of the network and the improvement of methods for observing avalanches, forecasting methods became more complicated and improved from those based on the intuition and experience of the forecaster to modern computer programs of artificial neural networks [4,18–20].

In the 1980s, the Swiss Snow and Avalanche Research Institute (SLF) developed the first avalanche forecast computer program based on the nearest neighborhood method [21,22]. The program was not designed to replace a forecaster but to help them in the process of analyzing the situation and making a decision. It selected the most similar events to the current or predicted situation from the entire set of previous events, showing what the conditions and consequences were. Currently, modern NXD2000 and NXD-REG versions of this program are used by the Swiss Avalanche Warning Service for local and regional avalanche danger forecasts and the daily avalanche bulletin [23]. The program has become widespread and is used in many avalanche centers in Europe and North America [9,10,19,23–27].

In the 1990s, artificial intelligence methods began to be used for forecasting avalanche danger [28–32]. This became possible thanks to the development of computer technology and the computing abilities of personal computers. Artificial neural networks simulate the work of a human brain and show good results in practical application for predicting the avalanche danger in different regions [28–32]. They are also used to assess avalanche hazard [33] and glacier mass balance [34].

The development of any avalanche forecasting method proceeds according to the following scheme: (1) the collection of the data on avalanche conditions and avalanche activity for the previous period; (2) establishing links between the indicators of the conditions for the formation of avalanches and the characteristics of avalanche activity; And (3) the use of these relationships to determine the level of avalanche danger under current or forecast weather and snow conditions. As indicators of the conditions for the formation of avalanches, air temperature, precipitation, as well as wind speed and direction are usually chosen among meteorological indicators, and snow depth and the presence of a weak layer among snow conditions. The indicators of avalanche activity are the size and distribution of avalanches, which can be summed up in the level of avalanche danger. Often, in addition to the parameters directly measured at observation points, calculated indicators received from actual data are used as input variables, for example, the air temperature trend or the rate of new snow settling. Sometimes snow cover indicators are modeled by physically driven models based on meteorological observations; for example, the SNOWPACK model in Switzerland [5,31,35] and SAFRAN/Crocus/MEPRA chain of models in France [8,36] are used for this.

In the initial period of the development of avalanche danger forecasts, an alternative approach was used, in which all situations were divided into two categories: "there are avalanches" or "there are no avalanches" [3,37,38]. In the 1990s, the probabilistic forecast of avalanches began to prevail, when the probability of avalanches and the intensity of the avalanche activity began to be assessed. In 1983, a five-level scale of avalanche danger was developed at the SLF [18,39], which, with minor modifications, began to be used in many countries in Europe and North America [40].

In the countries of the former USSR and in particular, in Kazakhstan, machine-based methods are very little used for avalanche forecasting, and the probabilistic forecast and the scale of avalanche danger levels are not used by official avalanche warning services [14].

Avalanche forecasts is an important part of avalanche prevention measures in the Almaty region. Since 1966, avalanche forecasting has been carried out by the avalanche department of the State Meteorological Service, which was created after the avalanche disaster in March 1966 [15]. In the same year, two snow avalanche stations began to operate in the Ile Alatau, where observations of the weather, snow cover, and avalanches were made. On the basis of these stations' work, methods for forecasting avalanches based on the discriminant analysis were developed [15,41,42].

The forecasts of new snow avalanches are carried out according to the method proposed by Tsomaya and Abdushelishvili in the Caucasus [43]. The forecast uses the data on the depth of old snow and the depth or water equivalent to new snow. On the scatter plot, points "with avalanches" and "without avalanches" are indicated. A dividing line is drawn between them and the corresponding equation is selected [44]. Kondrashov further improved this method and began to divide avalanche situations according to the size of avalanches [41]. Kolesnikov developed a method for forecasting avalanches associated with thaws, in which the critical sum of hourly positive air temperatures at the representative meteorological station was determined depending on the strength of old snow [15].

In 2000, within the framework of cooperation with the SLF, an attempt to use the NHD2000 program for forecasting avalanches in the valley of the Kishi Almaty River based on the long-term data (1966–1999) of the Shymbulak avalanche station was made [45]. The accuracy of the forecasts was 70%. Unfortunately, this experiment did not continue.

Currently, the Avalanche Warning Service of Kazakhstan uses two methods to predict avalanches [14]. The Kondrashov method is used for the forecast of avalanches associated with snowfalls. The Kolesnikov method is used for the forecast of avalanches associated with thaws. The main predictor for the forecast of snowfall avalanches is the amount of new snow and for the forecast of thaw avalanches, the maximum air temperature [15]. For all types of avalanches, the depth of the old snow is of great importance.

After the onset of a snowfall or thaw, measurements of the new snow depth or air temperature are carried out hourly, and when they approach critical values, an avalanche danger is declared. In this case, the regional office of the meteorological service sends a warning to the local administration and emergency services, which, if necessary, take protective measures [15].

As a matter of fact, this is not quite a forecast but rather a diagnosis of the avalanche danger, since a warning is practically issued when the avalanche danger has already occurred. Sometimes, this warning is issued when avalanches have already begun to occur. The forecast is considered correct if at least one avalanche occurred during the declared avalanche dangerous period. With such an approach, the accuracy of the currently used methods is equal to 90–95% [14]. At the same time, it should be noted that such a high level of correct forecasts is due not so much to the accuracy of the methods and forecast graphs but to the experience and intuition of forecasters.

The main advantage of these methods is simplicity. The disadvantages include the following: (1) no lead time; (2) the alternative nature of the forecast; (3) a strong dependence on the experience of the forecaster.

The main conclusions driven from the analysis of the current state of the forecast and warning of the avalanche danger in Kazakhstan are as follows. The methods were developed more than 40 years ago; they are largely outdated and require modernization. To improve forecasting methods, it is necessary to use the observational data from avalanche stations obtained over the past 20 years. When developing methods for forecasting avalanches, it is necessary to use new approaches and methods, in particular, machine learning methods and the international scale of avalanche danger levels.

Solving these problems was the aim of this work. The following tasks were complete. An electronic database of the weather, snow cover, and avalanches for the last 22 years was created. A five-level scale of avalanche danger adapted to the conditions of the Ile Alatau was developed. An artificial neural network was created. It was trained to assess and predict the levels of the avalanche danger.

The novelty of this research is the development of a method for forecasting avalanche danger using artificial neural networks and the international scale of avalanche danger in relation to the conditions of the Ile Alatau Ridge. The results of this work are of great importance for the improvement of avalanche forecasting methods in the countries of Central Asia.

## 2. Study Area

The study area covered the Kishi Almaty and Ulken Almaty River basins located on the northern slope of the Ile Alatau Ridge (Figure 1). This ridge is located in the southeast of Kazakhstan. It belongs to the Northern Tien Shan mountain system. The study area was limited by the coordinates from 43°00′ to 43°14′ N and from 76°56′ to 77°02′ E. The territory was chosen as a study area because it is characterized by a high avalanche activity and development. Therefore, avalanches are a big problem there. Over the past 55 years, 143 people have been caught up by avalanches, 67 of them have died. In total, 90% of the dead were skiers, tourists, and climbers [2,15].

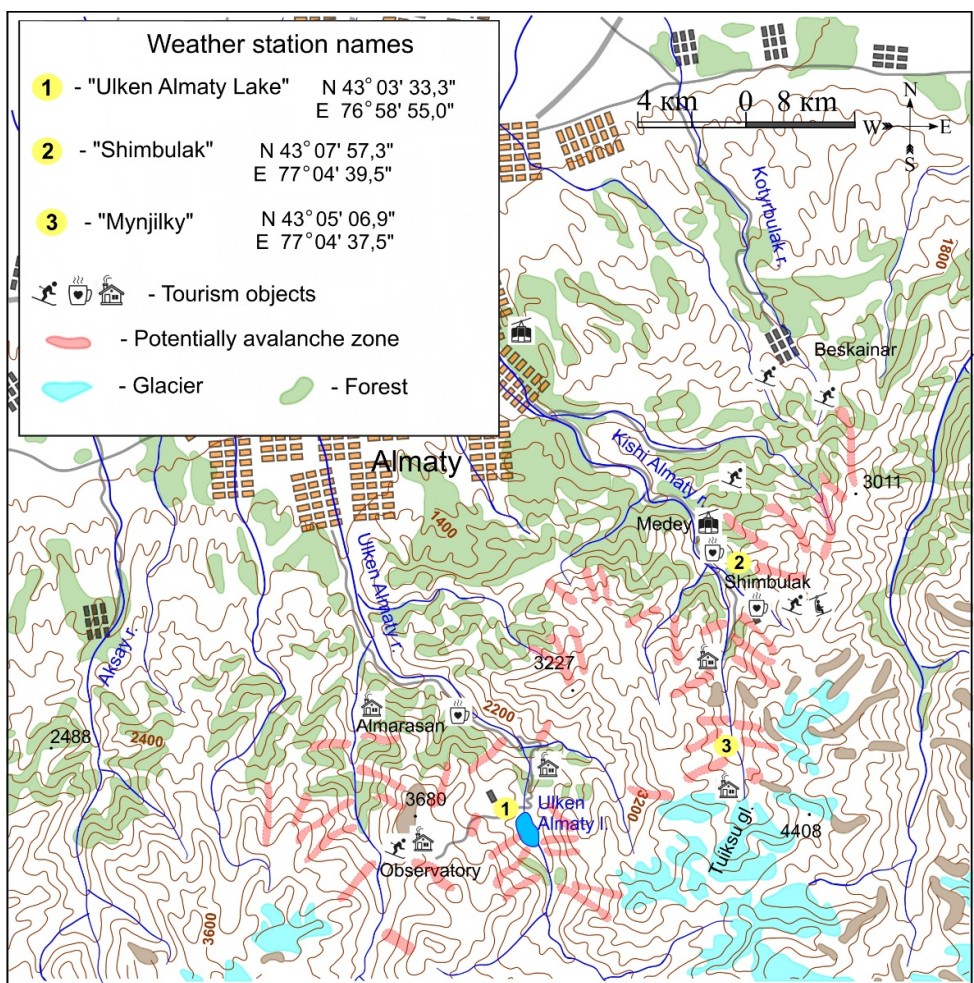

**Figure 1.** Study area.

Almaty, the largest city of Kazakhstan, is located at the foot of the mountains. Its population is about 2 million people. In the mountains, within an hour's reach, there is the large Shymbulak ski resort, which is visited by up to 12 thousand people daily. The area is very popular among climbers, freeriders, and ski tourists. A good knowledge of the conditions of avalanche formation and avalanche activity is of great importance [46,47]. There are three meteorological and two avalanche stations with a long history of observations.

The foot of the mountains is located at an altitude of about 1100 m above sea level. The height of the watershed ridges reaches 4400 m. Four high-altitude landscape zones are distinguished: a low-mountain one with deciduous forests and shrubs (1100–1500 m), a mid-mountain forest-meadow one with coniferous forests and subalpine meadows (1500–2800 m), a high-mountain meadow one with alpine meadows and creeping shrubs (2800–3400 m), and a high-mountain glacial one with stones, rocks, and glaciers (above 3400 m).

A cold period with air temperatures below 0 °C and a constant snow cover lasts from December to February in the low-mountain zone, from November to April in the midmountain zone, and from October to May in the high-mountain zone. The yearly amount of precipitation is 800 mm in the low-mountain zone. With altitude, the amount of precipitation increases and reaches 1100 mm in the high-mountain zone. The amount of cold period precipitation in the form of snow is 100 mm in the low-mountain zone, 300 mm in the midmountain zone, and 800 mm in the high-mountain zone. The depth of the snow cover before the end of the winter is 50 cm in the low-mountain zone, 110 cm in the midmountain zone, and 150 cm in the high-mountain zone [47].

Avalanche activity is noted above 1500 m and reaches its maximum in the zone of 2000–3200 m. This zone is characterized by the predominance of a steep relief. The excess of watersheds over the bottom of the valleys reaches 1200 m. The area of avalanche starting zones varies from 1 to 50 ha.

According to Armstrong's classification, the snowy climate of the Ile Alatau is continental [48]. The average temperature of the cold period is −6.8 °C, the depth of the snow cover is 106 cm [49,50]. The dangerous avalanche period lasts from December to April. Due to a shallow snow depth and low air temperatures, a weak layer of snow with a depth hoar is usually formed by early January in the lower part of the snow cover and reaches a thickness of 30 cm by early March [47]. The surface hoar is formed very rarely and quickly disappears in the continental climate conditions due to the low air humidity [51].

In the Ile Alatau, the main causes of avalanching are snowfalls and thaws. Dry avalanches of new snow, caused by snowfalls, predominate in quantity, and wet avalanches of old snow, associated with thaws, predominate in size.

During the winter period, 3 to 48 days with avalanches occur. There are two peaks of avalanche activity during the winter. The first, a weak one, is in December and the second, a strong one, is in March. The duration of avalanche cycles in December–February, when dry avalanches with new snow occur, is 2–3 days. In March–April, when wet avalanches associated with thaws occur, the duration of avalanche cycles depends on the duration of the thaw and can reach up to 7 days [51].

All indicators of avalanche activity strongly depend on the snowiness of the winter [49]. Both snowiness and avalanche activity in the Ile Alatau are characterized by strong interannual variability. In years with little snow, the snow depth is less than 70 cm, the number of days with avalanches is three, the number of avalanches is 10, and the total volume of avalanches is 5000 $m^3$. In snowy years, the snow depth exceeds 150 cm, the number of days with avalanches is 50, the number of avalanches is 240, and the total volume of avalanches is 2 million $m^3$ [49]. In 57 years of observations, 1965/1966 was the most extreme year in terms of snowiness and avalanche activity. The repeatability of such indicators is less than one time in 100 years [51].

## 3. Materials and Methods

This work used data from the Shymbulak, Big Almaty Lake, and Mynzhylki meteorological stations, as well as data from the Shymbulak and Big Almaty Lake avalanche stations. The duration of the observations was 22 years, from 2001 to 2022. The location of the stations is shown in Figure 1. The Shymbulak and Mynzhylki stations are located in the Kishi Almaty River valley at an altitude of 2200 and 3017 m, respectively. The Big Almaty Lake stations are located in the neighboring Ulken Almaty River valley at an altitude of 2502 m. The meteorological stations are a part of the World Meteorological Organization network, and the data of these stations are available on the website www.rp5.ru (accessed on 30 May 2022) [52]. Avalanche stations' data were taken from the Avalanche Warning Service reports. The work also used the data from observations of the snow cover and avalanches from the Institute of Geography and Water Safety.

The list of variables included in the database is given in Table 1. Meteorological variables were taken from the meteorological stations' data. The measurements of parameters at these stations were carried out according to the WMO and KazHydroMet standards [53,54].

The snow cover characteristics were taken from the reports of the avalanche stations. Observations at these stations were carried out in accordance with the Guidelines for Snow Avalanche Operations of Kazakhstan [55]. The snow depth was measured by stakes with an accuracy of 1 cm. The snow water equivalent was measured by a cylindrical-weight densitometer with an accuracy of 1 mm for the water layer. The snow depth and the snow water equivalent at the stations were measured daily. The snow depth on slopes was measured weekly and after snowfalls.

**Table 1.** The list of the database variables.

| Parameter | Units | Obtaining Way |
|---|---|---|
| Snow depth at mountain slopes | cm | Measured remotely |
| Snow depth at meteorological stations | cm | Measured at the site of a snow-avalanche station |
| New snow depth | cm | Measured at the site of a snow-avalanche station |
| Snowfall intensity | cm/hour | Calculated |
| Water equivalent of the snow cover | mm | Measured at the site of a snow-avalanche station |
| Presence of a weak layer in the snow cover | 2 categories: yes, no | Determined at the site of a snow-avalanche station |
| Snow shear strength in the weakest layer | kg/m$^2$ | Measured at the site of a snow-avalanche station |
| Snow water equivalent above a weak layer | mm | Measured at the site of a snow-avalanche station СЈС |
| Snow cover stability coefficient | No | Calculated |
| Snow cover stability index by the block test | 3 categories: stable, unstable, very unstable | Measured at the site of a snow-avalanche station |
| Daily amount of precipitation | mm | Measured at the meteorological site |
| Amount of precipitation per snowfall | mm | Calculated |
| Precipitation intensity | mm/hour | Calculated |
| Sum of precipitation for the previous 3 days | mm | Calculated |
| Average daily air temperature | °C | Calculated |
| Maximum air temperature | °C | Measured at the meteorological site |
| Minimum air temperature | °C | Measured at the meteorological site |
| Sum of hourly air temperatures since the beginning of the thaw | °C | Calculated |
| Sum of the maximum air temperatures for the previous 3 days | °C | Calculated |
| Maximum wind speed | m/s | Measured at the meteorological site |
| Number of avalanches | No | Visually calculated for the studied area |
| Avalanche size | 5 categories: small, medium, large, very large, extremely large | Visually determined |
| Presence of avalanche danger | 2 categories: yes, no | Assessed by a snow-avalanche station forecaster |
| Avalanche danger level | 5 categories: low, moderate, considerable, high, extreme | Assessed by avalanche experts |

The description of snow profiles was carried out according to the methods of the European, Canadian, and American avalanche warning services [56]. The shear strength of snow in a weak layer was measured using a hand-held dynamometer and a frame with an area of 100 cm$^2$ (Figure 2). The snow cover stability coefficient was calculated as the ratio of the snow strength to the water equivalent of the snow lying above the weak layer. Since 2019, the results of snow compression tests conducted according to the methodology of the Canadian Avalanche Association [56] have been used as an additional indicator of the snow cover stability.

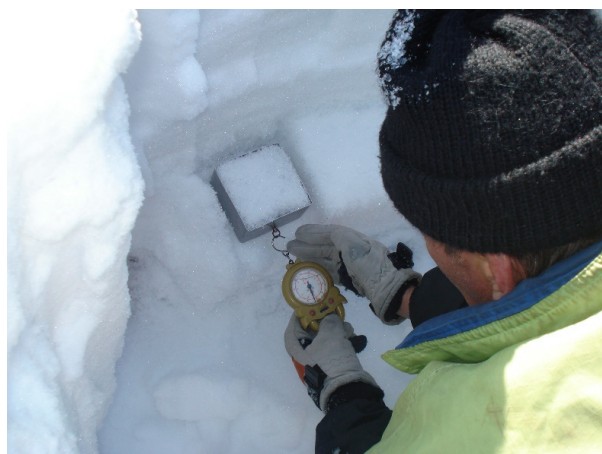

**Figure 2.** Measuring the shear strength of snow in a weak layer.

To characterize the avalanche activity, the data on the size and number of avalanches, based on the results of observations by the avalanche stations and the Institute of Geography and Water Safety was used. The types and reasons of avalanches were determined by the avalanche deposits and were controlled by weather conditions. The sizes of some avalanches were measured instrumentally by geodetic methods, but in most cases, they were visually assessed.

Avalanche sizes were divided into five categories according to the Canadian classification [56], used in Canada and the USA: (1) small, less than 0.1 thousand m$^3$, (2) medium, 0.1–1 thousand m$^3$, (3) large, 1–10 thousand m$^3$, (4) very large, 10–100 thousand m$^3$, and (5) extremely large, more than 100 thousand m$^3$. Based on these data, the avalanche dander level (ADL) was reconstructed for the period up to 2019, when the ADL had not yet been determined. Since 2019, the ADL has been assessed daily by the Institute of Geography and Water Safety specialists when compiling an avalanche bulletin.

To assess the ADL, a 5-level scale developed on the basis of the European and North American scales [39,40] was used, taking into account the conditions of the Ile Alatau (Table 2). Depending on the size and prevalence of avalanches, the following levels were distinguished: low, moderate, considerable, high, and extreme levels of avalanche danger.

**Table 2.** The scale of avalanche danger levels.

| Avalanche Danger Level | Size of Avalanches | Number of Avalanches | Probability of Human Triggering | Recommendations for Tourists | Protective Measures |
|---|---|---|---|---|---|
| 5 Extreme | Very large and extremely large | Numerous | Very high | Do not go to the mountains | Closure of roads and territories. Evacuation of people from the avalanche zone |
| 4 High | Very large | Numerous | Very high | Do not enter avalanche affected areas | Closure of roads and territories. Preventive avalanching |
| 3 Considerable | Large | Many | High | Choose the route carefully. Check snow stability | Warning of the population. Preventive avalanching in especially dangerous areas |
| 2 Moderate | Medium | Several | Low | Be careful on the slopes with specific snow conditions | Warning of the population |
| 1 Low | Small | Single | Very low | Do not go on snowy slopes steeper than 40 degrees | Informing the population |

The total number of daily observations of the weather, the snow cover and the avalanche danger level was 3960. The data of 3098 avalanches were used in the work. The timescale of the output variable was daily.

*Creation of the artificial neural network.* An artificial neural network can be created in two ways: writing an independent program in open-source software (for example, in the Python language) or using ready-made neurosimulators from computer program manufacturers, for example, MathCad or Statistica StatSoft.

In our work, we used the Perm State University neurosimulator [57] and the statistical software package from Statistica StatSoft Russia [58]. Tests of these programs showed that the Statistica StatSoft neurosimulator gave the best results. The Statistica package is a specialized software used in statistical and analytical work. It includes basic statistical tables, automated neural networks, and data mining. Basic statistical tables allow to statistically analyze data. The Statistica StatSoft neural network block allows to train a neural network to solve problems of classification, regression, and cluster analysis. It includes learning algorithms which give good results in supervised learning [58].

When training neural networks, two types of algorithms are used—supervised and unsupervised learning. In our case, the archive information contained an output variable (the avalanche danger level). Therefore, we chose a supervised learning algorithm. When working with the StatSoft neural network simulators, the iterative numerical optimization algorithm (BFGS) usually gives better results than the backpropagation algorithm [58].

A schematic diagram of the neural network is shown in Figure 3.

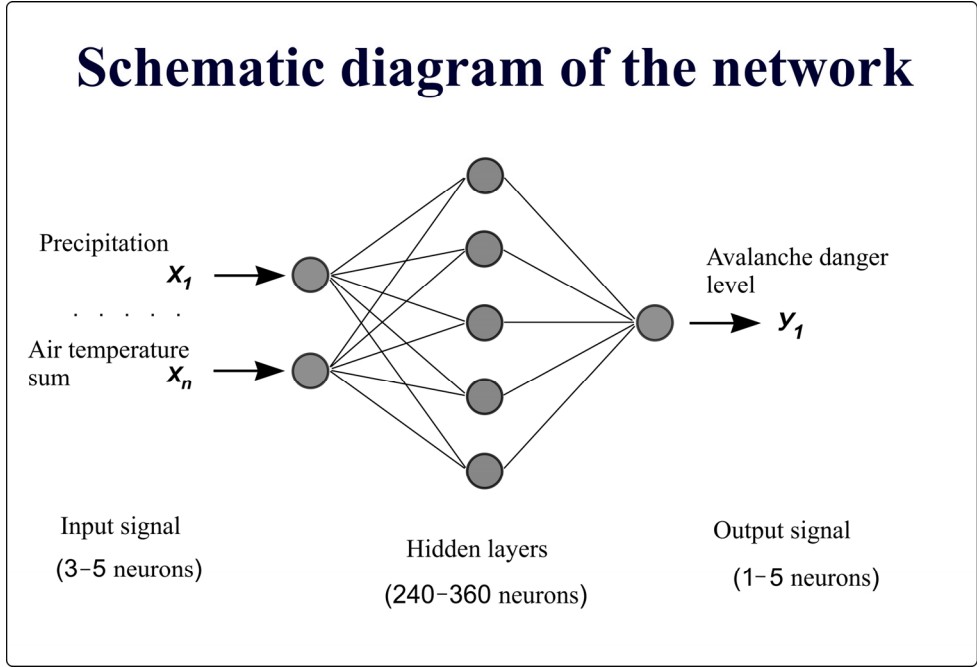

**Figure 3.** A schematic diagram of the artificial neural network.

The process of creating a computer model for assessing the avalanche danger level using an artificial neural network is schematically shown in Figure 4. It includes the following steps: collecting statistical data, choosing a mathematical calculation function, and testing the finished model. With unsatisfactory results, the steps are repeated.

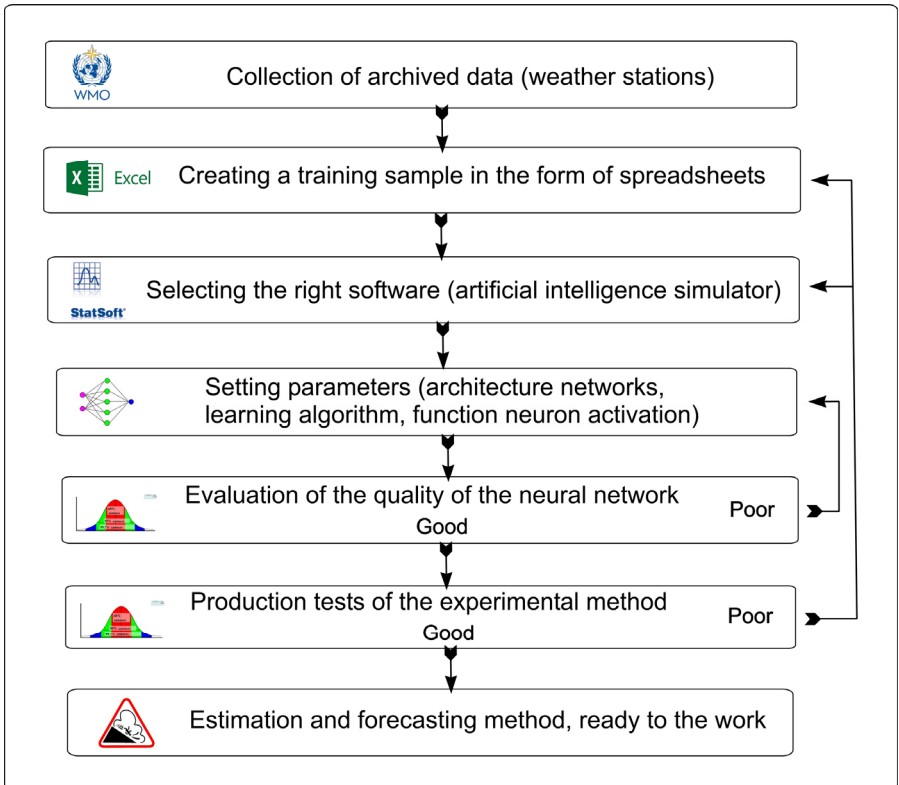

**Figure 4.** The stages of creating the avalanche forecasting method.

The archived weather and avalanche data were exported to a Statistica Spreadsheet and the training process was performed using the advanced automated neural network function. We used a neural network with the following parameters:

1. Network type: multilayer perceptron.
2. Statistical problem: regression and classification.
3. Number of hidden layers: 3.
4. Number of learning epochs: 3000.
5. Number of hidden neurons: 300.
6. Learning algorithm: iterative numerical optimization (BFGS).
7. Activation function of hidden neurons: hyperbolic.
8. Activation function of output neurons: identical.
9. Sampling division: 90% training, 5% validation, 5% test.

The numbers of hidden neurons (NHN) and training epochs (NTE) were chosen empirically. For this, several networks with NHNs from 240 to 360 and NTEs from 1000 to 3200 were tested. The networks were trained on the training set. The quality of the model was determined by the ratio of correctly estimated daily levels of avalanche danger to the total number of days in the sample set using a cross-validation. At the minimum values of the number of hyperparameters, the rate of correct estimation of the avalanche danger level was 77%. With their increase, the accuracy of the model increased to 90% when NHN = 300 and NTE = 3000. A further increase in the number of hyperparameters did not lead to an improvement in the quality of the model, so these values were chosen.

The training set was chosen as 90% of the data set in order to include more cases with high and extreme avalanche danger levels. The training set included data from the period 2001–2019, the validation set, data from 2020, and the test set, data from 2021.

The remaining neural network parameters were chosen based on the recommendations of the program manufacturer [58].

## 4. Results

The database on the weather, snow cover, and avalanches for 2001–2022 allowed us to obtain the following results: the frequency of avalanches caused by different reasons, the probability of avalanche cycle durations, the frequency of days with different avalanche danger levels and risks, the influence of factors on the avalanche danger level, the average and threshold values of avalanche formation factors for different avalanche danger levels, an artificial neural network trained to recognize avalanche danger levels, and a method for assessing the current and forecasting avalanche danger levels.

*Frequency of avalanches caused by different reasons.* Precipitation was the most common cause of avalanches (Figure 5). In 42% of cases, avalanches occurred either during a snowfall or 1–2 days after it. Such avalanches occurred in November–February. In 25% of cases, avalanches occurring during thaws were accompanied by precipitation. In those cases, the precipitation could be both in the form of snow or rain. The thaws without precipitation were the cause of avalanches in 26% of cases. The avalanches connected with thaws or thaws with precipitation occurred in March–April. These avalanches were the largest and most destructive ones. Blizzard and artificial avalanches accounted for 7% of cases. Artificial avalanches could be caused by skiers or explosions.

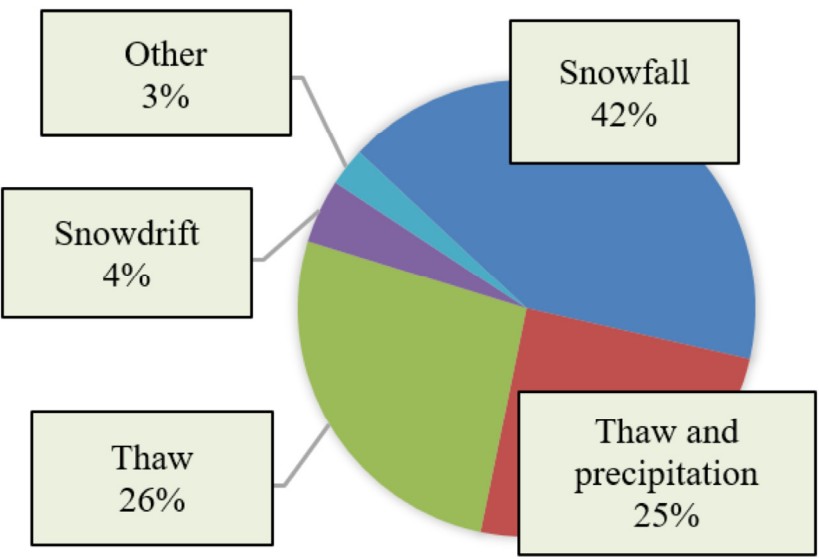

**Figure 5.** The reasons for avalanches.

*Duration of avalanche cycles.* Most often, avalanche cycles lasted 2 or 3 days. The occurrence of such cycles was 72% of the total number of avalanche cycles. Cycles when avalanches occurred 4 days in a row accounted for 14% of cases, and cycles when avalanches occurred from 5 to 7 days in a row accounted for 13% of cases.

*Frequency of days with different avalanche danger levels.* The distribution of the number of days with different avalanche danger levels is shown in Figure 6. During most of the cold period, low and moderate danger levels prevailed. At that time, single avalanches of small and medium sizes were recorded. They could threaten climbers and tourists. A considerable danger level was observed on 10.5% of days, mainly in the spring months. The number of days with high and extreme danger levels was only 1.9%.

Low and moderate levels of avalanche danger suggested a low probability of fatalities caused by avalanches. However, since such danger levels continued for a very long time (84% of the duration of the winter period), the proportion of deaths attributable to these danger levels was quite significant and amounted to 65%. In total, 12% of deaths occurred on the days with a considerable avalanche danger level, and 23% of such cases occurred on the days with a high and extreme danger levels. Taking into account the length of periods with different avalanche danger levels, an individual avalanche risk at a considerable danger level was 2.5 times higher than at low and moderate ones, and at high and extreme

levels, 7.5 times higher than at a considerable one. The probability of death on days with high and extreme avalanche danger levels was 18 times higher than the risk on days with a low and moderate danger.

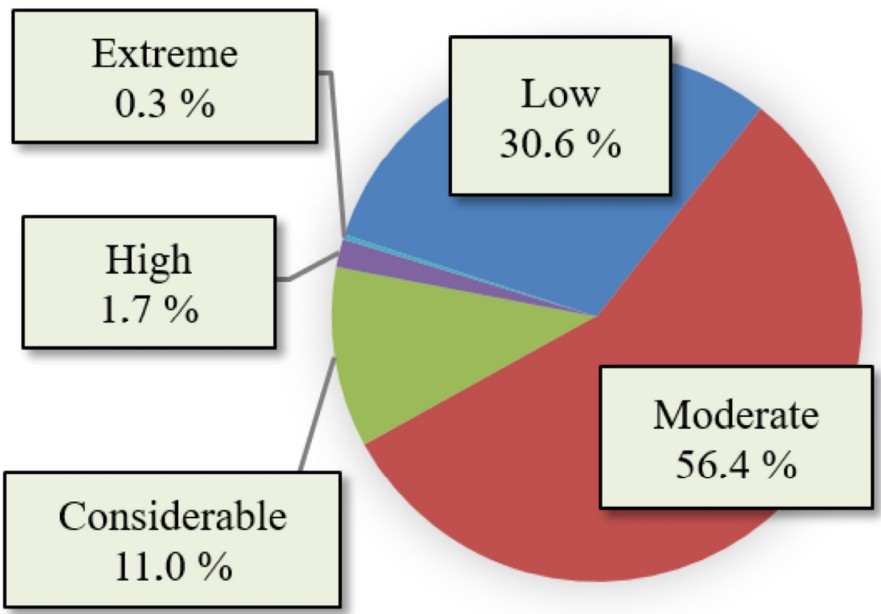

**Figure 6.** Frequency of days with different avalanche danger levels.

Avalanches with property damage occurred almost exclusively at high or extreme levels of avalanche danger, when they hit objects unprotected by engineering structures. These avalanches accounted for 80% of cases. Material damage from avalanches with a considerable danger level happened in 20% of cases. The risk of material damage from avalanches at high and extreme levels of avalanche danger was almost 150 times higher than at all other danger levels.

*The influence of the factors on the avalanche danger level* was assessed using the Spearman correlation coefficient (Table 3). Most of the variables showed a stronger correlation with the avalanche danger level than with the avalanche event. The highest correlation was observed between the avalanche danger level and the depth of old and new snow and between the avalanche event and the depth of new snow and the precipitation rate.

**Table 3.** The Spearman's correlation coefficients of variables with the avalanche danger level and the avalanche event.

| Variables | Avalanche Danger Level | Avalanche Event |
|---|---|---|
| Snow depth | 0.62 | 0.20 |
| New snow depth | 0.50 | 0.55 |
| Snow water equivalent | 0.63 | 0.21 |
| Coefficient of snowpack stability | −0.24 | −0.11 |
| Presence of a weak snow layer | −0.25 | −0.15 |
| Daily precipitation | 0.30 | 0.28 |
| Sum of precipitation for the previous 3 days | 0.41 | 0.32 |
| Precipitation rate | 0.43 | 0.48 |
| Snowfall rate | 0.37 | 0.41 |
| Minimum air temperature | 0.16 | 0.11 |
| Maximum air temperature | 0.18 | 0.13 |
| Average air temperature | 0.18 | 0.13 |
| Sum of the maximum temperatures for the previous 3 days | 0.24 | 0.19 |

*Average and threshold values of the avalanche formation factors for different avalanche danger levels* were obtained by the cluster analysis using the StatSoft.13 software package. The cluster analysis was carried out to solve the following problems: grouping the avalanche dangerous periods into relatively homogeneous classes in accordance with five levels of avalanche danger; a standardization of avalanche danger levels' assessment by experts, a reduction of the subjectivity, and assistance to an inexperienced expert; the creation of a clustering algorithm that could be used in little-studied regions.

When setting up the Statistica.13 program, the algorithms recommended by the manufacturer were used, including clustering according to the Varda method with the choice of the distance between points according to the Manhattan method [58]. As a result, the average values of the factors of avalanche formation for the five levels of avalanche danger were obtained (Table 4).

**Table 4.** Average values of the meteorological parameters for different avalanche danger levels.

| Meteorological Parameter | Avalanche Danger Level | | | | |
|---|---|---|---|---|---|
| | Low | Moderate | Significant | High | Extreme |
| Daily precipitation, mm | 1.0 | 12.6 | 20.0 | 35.0 | 40.0 |
| Precipitation rate, mm/h | 0.0 | 0.5 | 1.2 | 1.5 | 2.0 |
| Maximum air temperature, °C | −6.4 | −1.5 | 3.1 | 7.3 | 12.7 |
| Sum of the maximum temperatures for the previous 3 days, °C | −16.5 | −1.2 | 10.9 | 23.2 | 36.9 |
| Snow depth, cm | 25 | 45 | 62 | 72 | 84 |
| Snow cover water equivalent, mm | 48 | 97 | 142 | 189 | 248 |
| Coefficient of snowpack stability | 1.55 | 1.15 | 0.99 | 0.84 | 0.82 |

The data on the threshold values of meteorological elements corresponding to three levels of avalanche danger are given in Table 5. The threshold values of factors between two and three and between four and five levels of avalanche danger could not be obtained due to a strong blurring of the boundaries between clusters.

**Table 5.** Threshold values of the snow cover characteristics and meteorological parameters for different avalanche danger levels.

| Snow Cover Characteristics and Meteorological Parameters | Avalanche Danger Level | | |
|---|---|---|---|
| | Low | Moderate and Significant | High and Extreme |
| Daily precipitation, mm | 10 | 15 | 25 |
| Precipitation rate, mm/hour | 0.5 | 1.0 | 1.2 |
| Maximum air temperature, °C | 10 | 15 | 20 |
| Sum of the maximum temperatures for the previous 3 days, °C | −1 | 5 | 20 |
| Snow depth, cm | 30 | 50 | 75 |
| Snow water equivalent, mm | 100 | 150 | 200 |
| Coefficient of snowpack stability | 1.5 | 1.0 | 0.7 |
| Depth of the snow cover on avalanche prone slopes, cm | 50 | 75 | 100 |
| New snow depth, cm | 15 | 20 | 30 |
| Snowfall rate, cm/hour | 1.0 | 1.5 | 2.0 |
| Water equivalent of a new snow, mm | 10 | 15 | 20 |
| Presence of a weak snow layer in the snowpack | No | Yes | Yes |
| Assessment of the stability of the snow cover by block tests' method | Stable | Unstable | Very unstable |
| Sum of hourly air temperatures since the beginning of the thaw, °C | 200 | 300 | 400 |
| Wind speed (gusts), m/s | 10 | 15 | 20 |

*The results of training the artificial neural network to recognize avalanche danger levels.* The input variables for training the artificial neural network were the following: the daily precipitation, the old snow depth, the new snow depth, the maximum air temperature, the sum of the three previous days' precipitations, the sum of the three previous days'

maximum air temperatures, and the index of snowpack stability. The output variable was the avalanche danger level.

While performing the work, several neural networks with different learning parameters (number of epochs, number of neurons, learning algorithm) were trained. The ones that showed the smallest error were selected among them. The finished neural networks were saved in the PMML format—the international standard for intelligent programs. They can be opened in the Statistica program in the prediction mode for further work while operationally forecasting avalanches.

A neural network simulator trained on the training sample gave good results (Table 6). In the regression mode, the percentages of recognition were 84–90%, and in the classification mode, they were 77–91%. The quality of the network was determined by the ratio of correctly estimated daily levels of avalanche danger to the total number of days in the sample set using cross-validation.

**Table 6.** Results of training and testing the neural network simulator from StatSoft to assess the avalanche danger level.

| Observation Station | Network Architecture | Network Quality (% of Recognition) | | |
| --- | --- | --- | --- | --- |
| | | Learning Performance | Test Sample | Validation Sample |
| Network operation in the regression mode | | | | |
| Shymbulak | MLP 5-240-1 | 86.3 | 87.2 | 86.3 |
| Big Almaty Lake | MLP 5-320-1 | 88.3 | 89.1 | 89.6 |
| Mynzhylki | MLP 3-240-1 | 85.5 | 89.6 | 84.1 |
| Network operation in the classification mode | | | | |
| Shymbulak | MLP 6-240-5 | 90.5 | 83.5 | 81.6 |
| Big Almaty Lake | MLP 6-260-5 | 88.7 | 84.8 | 85.4 |
| Mynzhylki | MLP 3-240-5 | 80.6 | 83.0 | 76.7 |

*Avalanche danger level forecast based on numerical weather forecast models.* The main drawback of the existing methods for forecasting avalanche danger is the lack of lead time. One way to increase the lead time is to use the results of numerical weather forecasting models. In that case, the accuracy of the avalanche danger level's estimation slightly decreased.

The reference points for the snow avalanche forecast were the three mountain meteorological stations—Shymbulak, Mynzhylki, and Big Almaty Lake. The weather forecast for these points can be taken from the interactive weather map on the Windy.com website [59]. We used the data from three predictive models: ECMWF, GFS, and ICON. The GFS model showed the best accuracy in the conditions of the Ile Alatau in terms of precipitation, and the ICON model in terms of air temperature. The values of the correlation coefficients for the air temperature were higher than for the amount of precipitation. As the terrain altitude rose, the accuracy of the weather simulation results decreased.

For avalanche danger levels' forecasts with a lead time of up to 3 days, the GFS model should be used for precipitation forecasts and the ICON model for air temperature forecasts. The accuracy of estimating the forecasted avalanche danger levels was reduced to 75–80% due to errors in weather forecasts.

In the winter of 2021/2022, the developed method was tested on data that were not included in the 2021–2021 set. With the help of a neural network trained on a sample of 2001–2021, the current (today's) and forecast (tomorrow's) levels of avalanche danger were estimated. They were compared with the actual levels of avalanche danger, which were determined by experts on the basis of data on avalanche activity. Out of 175 days, the artificial neural network correctly estimated the current level of avalanche danger in 157 cases (90%). The forecast danger levels coincided with the actual ones in 80% of cases.

## 5. Discussion

The values of the avalanche formation factors at different levels of avalanche danger were grouped into clusters with certain average and threshold values. The differences between the clusters were most clearly expressed for moderate and high avalanche danger levels. Thus, for a moderate level, the average value of daily precipitation was 12.6 mm, the sum of 3 days' maximum air temperatures was −1.2 °C, and the coefficient of snowpack stability was 1.15. For a high level, these values were 35.0 mm, 23.2 °C and 0.84, respectively. These results are of great help to a not very experienced avalanche expert when evaluating avalanche danger levels.

In the Ile Alatau, the main causes of avalanches were snowfalls and thaws. Avalanches caused by snowfalls accounted for 42% of the total number of avalanches, 26% of avalanches were caused only by thaws, and 23% of avalanches occurred during thaws accompanied by precipitation. The share of other types of avalanches (blizzard and artificial ones) accounted for only 7% of avalanches.

During the winter period, low and moderate levels of avalanche danger prevailed. The number of days with such avalanche danger levels averaged 82.4%. However, days with these levels of avalanche danger accounted for 68% of avalanche deaths. In total, 90% of such cases occurred with skiers, tourists, and climbers. A considerable level of avalanche danger was noted in 10.5% of cases. With such an avalanche danger level, 13% of avalanche incidents occurred. High and extreme levels of avalanche danger occurred in 1.9% of cases, 3 days during the winter period on average. With such avalanche danger levels, cases of death are rare, since emergency services close dangerous territories for this period. However, if we take into account the duration of periods with different levels of avalanche danger, it turns out that the individual avalanche risk on days with a considerable level of avalanche danger was 2.5 times higher than on days with a moderate level of danger. On days with a high level of avalanche danger, the individual avalanche risk was 7.5 times higher than on days with a considerable danger level.

The largest avalanches occurred at high and extreme levels of avalanche danger. They damaged insufficiently protected tourist infrastructure facilities. With such avalanche danger levels, 80% of cases of avalanches that caused material damage occurred. The risk of property damage from avalanches on days with high and very high levels of avalanche danger was almost 150 times higher than on days with low and moderate levels of avalanche danger.

The Artificial Neural Networks module of the StatSoft.13 software package could be successfully used in the operation of the avalanche warning service in Kazakhstan. The artificial neural network created with its help, trained on the data from 20 years of observations of precipitation, snow cover, and air temperature, was capable of correctly assessing the current level of avalanche danger in 85–90% of cases. For avalanche danger levels' forecasts with a lead time of up to 3 days, the accuracy was reduced to 75–80% due to errors in weather forecasts.

The artificial neural network has been successfully used by the Institute of Geography and Water Safety to assess and predict avalanche danger levels when compiling an avalanche bulletin. The bulletin is compiled for the most visited basins of the Kishi and Ulken Almaty Rivers and is distributed on a channel of the Telegram social network.

The accuracy of the local forecasts of avalanche danger without differentiation according to its levels, developed by the Hydrometeorological Service of Kazakhstan, is currently 85–95% [15], in Russia, 75–85% [60], and in India, 80–85% [61].

The percentage of correct danger level predictions for dry-snow conditions at the Switzerland Institute for Snow and Avalanche Research was 74–78% [32]. The precision for predicting the local avalanche danger level by the model developed at the Austrian Research Centre for Forests was 0.73 [31]. In Colorado Mountains, neural networks correctly predicted the avalanche activity in 78 to 91% of cases [28].

Prospects for the development of the method for assessing and forecasting avalanche danger in the Ile Alatau using artificial neural networks lie in the creation of a network

of automatic stations for the monitoring of the avalanche formation conditions. It is planned to install 10 automatic meteorological stations at the bottom of the valleys and 23 automatic snow posts on the slopes near avalanche starting zones. Meteorological stations will measure precipitation, snow water equivalent, air temperature and humidity, snow temperature, wind direction and speed, and solar radiation. At snow posts, instead of solar radiation and snow water equivalent, snowdrift will be measured. The monitoring data will increase the volume of information about the factors influencing avalanche formation and, as the data are accumulated, will allow to refine the forecast for individual avalanche sites or groups of similar sites and possibly to predict the avalanche time and the avalanche size.

**Author Contributions:** Conceptualization, A.M. and V.B.; methodology, V.B. and V.Z.; validation, V.Z.; formal analysis, V.B., T.G., V.Z. and S.R.; investigation, V.B., T.G., V.Z. and S.R.; data curation, A.K. and U.A.; writing—original draft preparation, A.M., V.B. and V.Z.; writing—review and editing, V.B.; supervision, A.M.; project administration, A.M. All authors have read and agreed to the published version of the manuscript.

**Funding:** This research was funded by the Ministry of Science and High Education of the Republic of Kazakhstan, grant number AP09260155.

**Data Availability Statement:** Can be obtained upon request.

**Conflicts of Interest:** The authors declare no conflict of interest. The funder had no role in the design of the study; in the collection, analyses, or interpretation of data; in the writing of the manuscript, or in the decision to publish the results.

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
