# Peer review of "Application of Artificial Intelligence in the Assessment and Forecast of Avalanche Danger in the Ile Alatau Ridge"

_water, doi:10.3390/w15071438_

Round 1

Reviewer 1 Report

The study is interesting and most importantly it has significant practical impact. The literature review is fine, though the methodology, presentation of results and discussion need to be improved. Please address the following comments:

1. L17 extra space needs to be deleted.

2. L17 accuracy – what was the metric of accuracy?

3.  L99-106 What are the novelties of this research?

4. L107 – please use a more representative section title.

5. Table 1 – snow depth in slopes and stations- is it at a daily level? Please clarify in the manuscript.

6. L227- Please state the number of observations and the timescale of the output variable (i.e., daily, monthly etc.?).

7. L 294-296 The choice of hyperparameters/options/layers of NN is not explained sufficiently. Please elaborate and clearly motivate these choices.

8. Figure 3 – low quality please improve.

9. L323 – usually we split into 70-30% but ok. Does control set mean validation set? Was the division on the basis of time (i.e., recent years are the hold out set)? Or completely randomly?

10. Figures 5-6 low quality

11. L408-410. Was this training on the training set (90%)? Was there any cross-validation? How were the different combinations of hyperparameters defined?

12. Table 6 – How is network quality defined mathematically?

13. Discussion – please comment on whether these results generalise, it would be nice to compare with other studies in other areas but with similar objectives.

Round 2

Reviewer 1 Report

The authors have adeuqately addressed all comments/suggestions. The workflow for the NN models development is much clearer now with justification of parameter selection.